# Hydrogeochemical Characterization and Its Seasonal Changes of Groundwater Based on Self-Organizing Maps

Chu Wu [1,*], Xiong Wu [2], Chuiyu Lu [1], Qingyan Sun [1], Xin He [1], Lingjia Yan [1] and Tao Qin [1]

1 State Key Laboratory of Water Cycle Simulation and Regulation, China Institute of Water Resources and Hydropower Research, Beijing 100038, China; cylu@iwhr.com (C.L.); sunqingyan123@163.com (Q.S.); hex@iwhr.com (X.H.); YLJ_cherry426@126.com (L.Y.); qtplayer@sina.com (T.Q.)
2 School of Water Resources and Environment, China University of Geosciences, Beijing 100083, China; wuxcugb@126.com
* Correspondence: wuchu@iwhr.com

**Abstract:** Water resources are scarce in arid or semiarid areas; groundwater is an important water source to maintain residents' lives and the social economy; and identifying the hydrogeochemical characteristics of groundwater and its seasonal changes is a prerequisite for sustainable use and protection of groundwater. This study takes the Hongjiannao Basin as an example, and the Piper diagram, the Gibbs diagram, the Gaillardet diagram, the Chlor-alkali index, the saturation index, and the ion ratio were used to analyze the hydrogeochemical characteristics of groundwater. Meanwhile, based on self-organizing maps (SOM), quantification error (QE), topological error (TE), and the K-means algorithm, groundwater chemical data analysis was carried out to explore its seasonal variability. The results show that (1) the formation of groundwater chemistry in the study area was controlled by water–rock interactions and cation exchange, and the hydrochemical facies were $HCO_3$-Ca type, $HCO_3$-Na type, and Cl-Na type. (2) Groundwater chemical composition was mainly controlled by silicate weathering and carbonate dissolution, and the dissolution of halite, gypsum, and fluorite dominated the contribution of ions, while most dolomite and calcite were in a precipitated state or were reactive minerals. (3) All groundwater samples in wet and dry seasons were divided into five clusters, and the hydrochemical facies of clusters 1, 2, and 3 were $HCO_3$-Ca type; cluster 4 was $HCO_3$-Na type; and cluster 5 was Cl-Na type. (4) Thirty samples changed in the same clusters, and the groundwater chemistry characteristics of nine samples showed obvious seasonal variability, while the seasonal changes of groundwater hydrogeochemical characteristics were not significant.

**Keywords:** self-organizing map; hydrogeochemistry; seasonal variation; cluster analysis; Hongjiannao

## 1. Introduction

Groundwater plays a crucial role in domestic and irrigation activities in arid–semiarid regions where surface water resources are short in supply or low in quality [1–3]. Rapid urban and industrial growth has led to overexploitation of groundwater, causing increasingly prominent water-related problems (e.g., depletion of water resources accompanied by groundwater pollution) in local areas [4,5]. Hydrogeochemical analysis of groundwater is an important aspect of hydrogeological research, as it guides the sustainable use and managementof groundwater resources as well as ecological and environmental protection [6–8]. Hence, there is an urgent need to determine the characteristics and seasonal variability of the hydrogeochemical composition of regional groundwater to guide the implementation of management measures for groundwater resources and prevent their further deterioration.

The Hongjiannao Lake Basin is located at the eastern margin of the Cretaceous basin on the Ordos Plateau. Over the past two decades, unreasonable development and utilization of water resources, as well as frequent anthropogenic engineering activities have reduced

the quality of water in the Hongjiannao Lake and caused ecological and environmental damage to the area [9–11]. Quaternary and Cretaceous groundwater discharge in the lake basin recharges and conserves Hongjiannao Lake. Groundwater is an important source of water that supports domestic and socioeconomic activities in this area [12,13]. Studies have tended to focus on the interpretation of the surface of the Hongjiannao Lake and its natural and anthropogenic influencing factors, as well as the flora and fauna resources in the Hongjiannao wetlands [14–16], while few have examined this area from a hydrogeological perspective [17]. No studies on the characteristics and seasonal variability of the hydrogeochemical composition of the groundwater in this lake basin have been reported.

Cluster analysis is a widely used, practical tool for studying the hydrogeochemical characteristics of groundwater [7,18]. Common clustering methods include principal component analysis, hierarchical cluster analysis, and k-means clustering. They can be employed to cluster samples of groundwater to characterize and assess its quality and to determine its hydrochemical characteristics [19–21]. Self-organizing maps (SOMs) can map complex high-dimensional data onto low-dimensional spaces to enable their visualization, based on the principle of competitive learning in artificial neural networks [22–24]. A SOM assigns samples with similar characteristics to one cluster while preserving their initial topological relationships. In other words, this technique places similar samples in one cluster and dissimilar samples in different clusters [25]. SOM-based clustering has been applied to the hydrogeochemical analysis and quality assessment of groundwater. Some researchers have noted that the selection of a suitable neuron size is the key to SOM-based clustering and have provided helpful theoretical guidance [20,21,26].

In this study, the hydrogeochemical characteristics of the groundwater in the Hongjiannao Lake Basin were determined by analyzing samples collected during the rainy and dry seasons using techniques such as Piper, Gibbs, and Gaillardet diagrams; the chloro-alkaline indices (CAIs); the saturation index (SI); and ion ratio analysis. The seasonal variability of the hydrogeochemical characteristics of the groundwater in the study area was determined through correlation and cluster analysis of its hydrochemical parameters using SOMs, the quantization error (QE), the topological error (TE), and k-means clustering. The research approach introduced in this study can provide theoretical and technical support for investigating the seasonal variability of the hydrogeochemical characteristics of groundwater in similar areas.

## 2. Description of the Study Area

### 2.1. Study Location and Climate

Encompassing an area of approximately 1440 km$^2$, the Hongjiannao Lake Basin lies at the eastern margin of the Maowusu Desert and the junction of Shaanxi and Inner Mongolia (Figure 1). Located in a relatively low-lying area of the Maowusu Desert, Hongjiannao Lake is the largest desert freshwater lake in China, with a surface area of approximately 38 km$^2$. The Hongjiannao Lake Basin has an arid–semiarid temperate plateau continental climate. Analysis of the variation in annual precipitation and evaporation in this area from 1990–2018 revealed the following. The multi-year annual average precipitation was 356.4 mm. Annually, most precipitation (69% of the total annual precipitation) fell in July through September (Figure 2), mainly in the form of rainstorms and with a maximum monthly precipitation of 223.7 mm, which tended to cause flooding. Evaporation peaked in April and then gradually decreased each month until October when it rebounded slightly. The multiyear maximum monthly evaporation was 258.1 mm, while the multiyear annual average evaporation was 1328.5 mm (measured with evaporating dishes with a diameter of 20 cm).

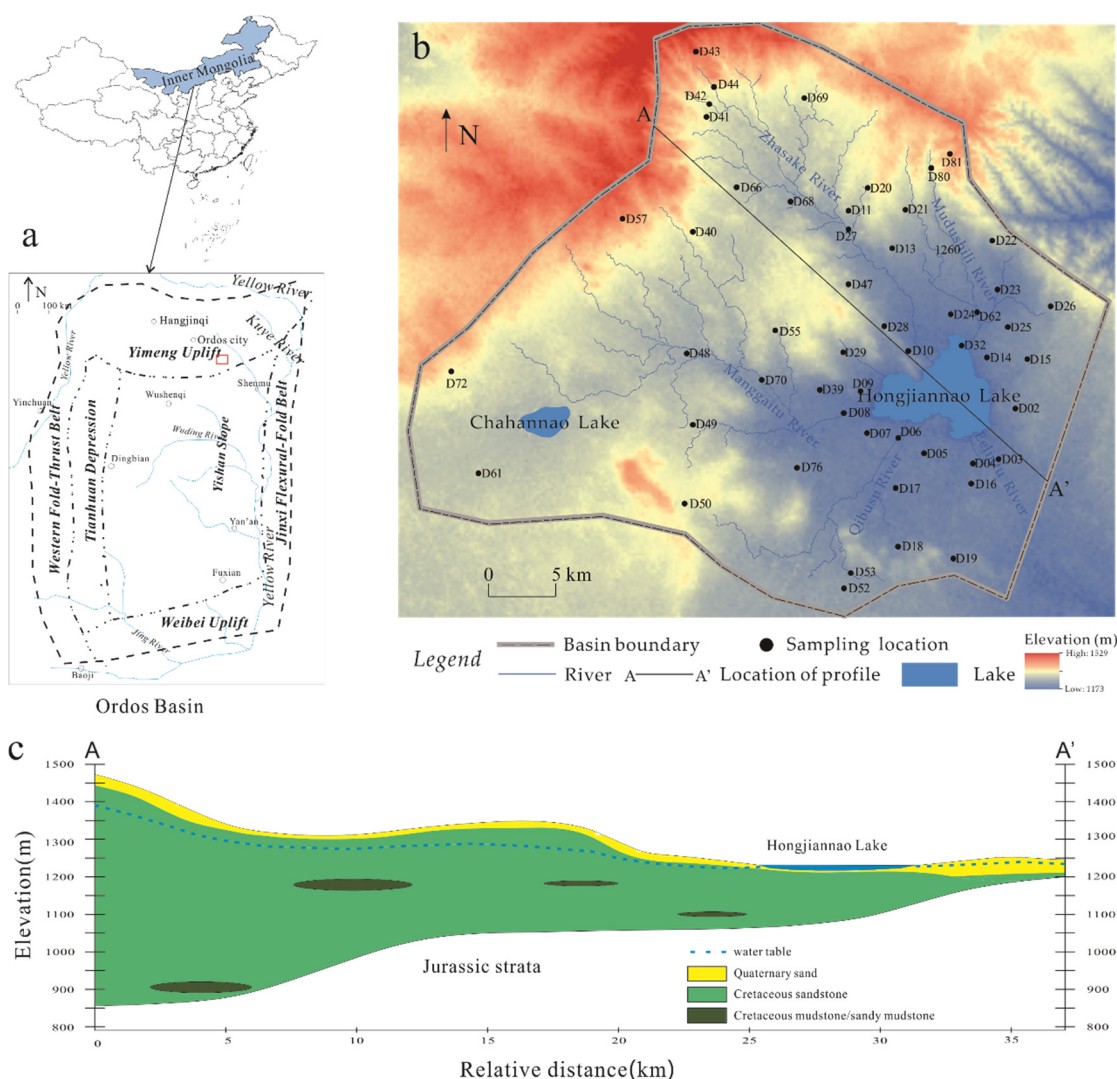

**Figure 1.** Location and hydrogeological condition of the study area and sampling locations ((**a**). Tectonic geology sketch of Ordos Basin (**b**). Sampling locations (**c**). Hydrogeological profile of A–A′.)

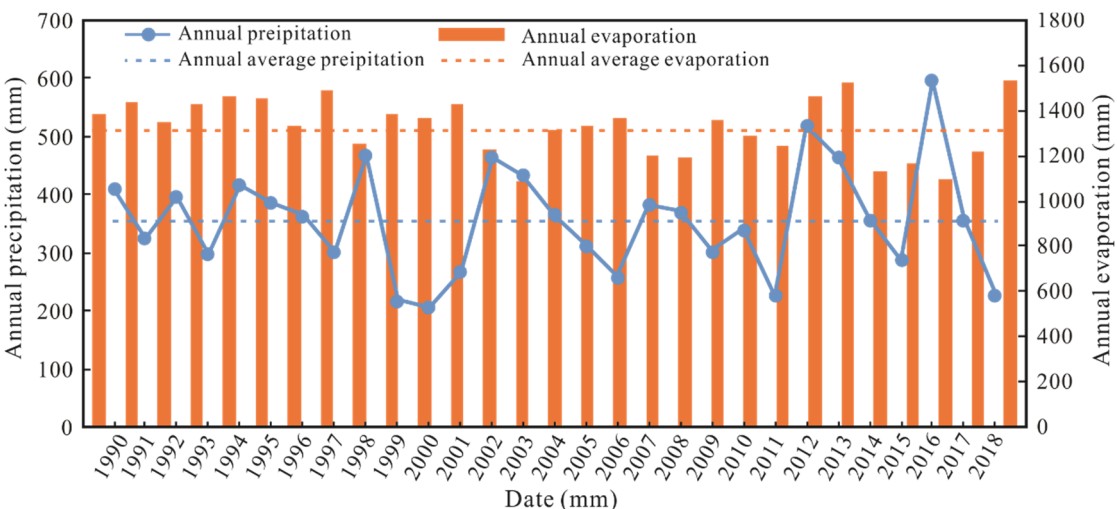

**Figure 2.** Variation of the annual precipitation and evaporation over a multiyear period.

## 2.2. Hydrogeological Setting

The distribution and characteristics of the groundwater in the Hongjiannao Lake Basin are controlled primarily by factors such as the terrain, landforms, lithology, geological structures, and climatic conditions. Based on the type of water-bearing rock, the groundwater in this area can be categorized into pore water in loose rocks and pore-fissure water (PFW) in clastic rocks. (1) There are three main loose rock formations that contain pore water. (i) There are the phreatic pore water (PPW)–bearing quaternary–Holocene alluvial and diluvial formations. Distributed in a wide-valley banded pattern in the riverbeds and terraces in the gullies across the lake basin, this formation contains PPW and has a lithology composed mainly of gravelly, medium- to fine-grained sand with gravel and pebble layers at the bottom. (ii) There are PPW-bearing quaternary–Holocene aeolian deposit formations. Distributed in the terraces and tableland beams on both sides of the gullies as well as at the margin of the Maowusu Desert, this formation has a lithological composition characterized by aeolian-deposited yellow, silty, fine-grained sand and medium- to fine-grained sand. (iii) There are PPW-bearing quaternary–Upper Pleistocene Salawusu formations. Distributed around the Hongjiannao Lake and in the southwestern part of the lake basin, this formation has a lithology consisting of light-yellow, silty, fine-grained sand as well as clay-like and sandy loam. (2) There are two main PFW-bearing Cretaceous clastic rock formations. (i) There is the Phreatic PFW (PPFW)–bearing lower Cretaceous Huanhe formation. This formation is distributed in the western and northern parts of the study area and the upper reaches of the Zhasake, Mudushili, and Manggaitu Rivers, with an aquifer composed lithologically of purplish-red and grayish-green sandstone, sandy gravel, and gravelly sandstone. (ii) There is the Phreatic PFW–bearing Lower Cretaceous Luohe formation. Distributed widely across the study area, this formation, together with the overlying quaternary aquifer, generally forms a uniform, nearly horizontally cross-bedded aquifer with a lithological composition consisting mainly of brownish-red medium- and fine-grained sandstone.

## 3. Data and Methods

### 3.1. Sample Collection and Analysis

In the study area, groundwater was sampled from different motor-pumped wells from August to September (the rainy season) in 2019 and March to April (the dry season) in 2020, collecting 42 rainy-season samples and 42 dry-season samples. Water samples from different seasons were taken from the same wells. A total of 84 samples were filtered with 0.45-μm filter membranes and collected in clean and dry polyethylene plastic bottles after pumping until the flowing water showed stabilized temperature, pH, dissolved-$O_2$, and Eh values. Sample collection, handling, and storage followed the standard procedures recommended by the Chinese Ministry of Water Resources [27]. The sampling locations were spaced as evenly as possible in the study area, as shown in Figure 1, where the field-based water parameters such as temperature, pH, and electric conductivity (EC) were measured in situ by HANNA portable instruments. Chemical and isotope analyses of water samples and sediment samples were performed at the Nuclear Industrial Geology Analysis and Testing Research Center, Beijing, China. The dissolved concentrations of major anions ($Cl^-$, $HCO_3^-$, $CO_3^{2-}$, and $SO_4^{2-}$) and cations ($Ca^{2+}$, $Mg^{2+}$, $K^+$, and $Na^+$) were analyzed using ion chromatography (ICS-1100 systems). The accuracy of the water quality testing was assessed using blank samples, parallel samples, and internal standards. The charge balance error percentage (%CBE) was calculated to be less than 5%, suggesting that the accuracy of each index met quality requirements.

### 3.2. Methods

First proposed by Kohen of the Helsinki University of Technology, Finland, in 1981 [23], SOMs are neural networks based on unsupervised learning. They have come to be extensively used in fields such as hydrology and environmental sciences. Researchers often apply SOMs to the cluster analysis of hydrogeochemical data [2,20]. In this study, we

examined the seasonal variability of the groundwater in the study area through a SOM-based cluster analysis of its hydrochemical parameters. The SOM-based clustering of groundwater samples collected from the study area consisted mainly of three steps: the selection of neurons, the selection of types, and the assignment of the samples to different types.

Selecting suitable neurons is the key to producing good clustering results. Generally, two metrics—QE and TE—can be used to evaluate the quality of the selected network size [21] and, on this basis, determine the optimal number of mapping neurons. The number of neurons in a neural network and the side lengths of its rectangle are determined through the minimization of TE and QE within their respective possible ranges. To determine the optimal number of matching neurons, Nguyen et al. [20] used the following empirical equation: $m = 5\sqrt{n}$, where $m$ is the number of neurons in the SOM, and $n$ is the number of samples input into the SOM. In this study, a total of 84 groundwater samples were collected from the study area during the rainy and dry seasons. Thus, $m = 5\sqrt{84} \approx 46$.

To better display the temporal and spatial distribution patterns of the hydrogeochemical characteristics of the groundwater in the study area, the 84 groundwater samples collected during the rainy and dry seasons were simultaneously used as input samples. Eight hydrochemical parameters, namely, the concentrations of $Na^+$, $K^+$, $Ca^{2+}$, $Mg^{2+}$, $Cl^-$, $SO_4^{2-}$, $HCO_3^-$, and $CO_3^{2-}$, were used in the cluster analysis. The SOM algorithms were trained on the data of the groundwater samples. TE and QE were calculated. With $7 \times 7 = 49$ neurons, TE = 0.0119, and QE = 1.0463, both of which were the minimum values within their respective ranges. Based on the results yielded by the two methods used to determine the number of SOM neurons, the number of matching neurons was optimized in this study to 49.

The Davies–Bouldin Index (DBI) is a metric for evaluating the quality of clustering algorithms [28]. For m time series that can be grouped into n clusters, let us set the $m$ time series as the input matrix $X$ and the n clusters as $N$, which is passed into the algorithm as a parameter. The *DBI* is calculated using Equation (1) as follows:

$$DBI = \frac{1}{N}\sum_{i=1}^{N}\max_{j \neq i}\left(\frac{\overline{S_i} + \overline{S_j}}{\left\| w_i - w_j \right\|_2}\right) \tag{1}$$

A small *DBI* value indicates that the data points within the same cluster are close to each other and that different clusters are far apart. That is, the minimum *DBI* value corresponds to an optimal number of clusters, $N_c$.

## 4. Results and Discussion

### 4.1. Hydrochemical Characteristics of Groundwater

Table 1 summarizes the hydrochemical parameters of the groundwater in the study area during the rainy season. The concentration of $K^+$, a basic element required for human health, was overall very low in the groundwater [3], ranging from 0.38 to 9.73 mg/L and averaging at 1.66 mg/L. Overall, there was a strong correlation between the concentrations of $Na^+$ and $Cl^-$ in the groundwater. The average concentration (76.18 mg/L) of $Na^+$ was greater than that of $Cl^-$ (40.60 mg/L). Both the average concentrations of $Na^+$ and $Cl^-$ were below their limits (200 and 250 mg/L, respectively) stipulated in the National Drinking Water Standards [27]. That the concentration of $Na^+$ was higher than that of $Cl^-$ in the groundwater may be attributed to the dissolution or cation-exchange reactions of other Na-bearing minerals in the groundwater environment [19,29].

**Table 1.** Statistics of the hydrochemical parameters of the groundwater in the study area during the rainy season (unit: mg/L).

|  | $Na^+$ | $K^+$ | $Mg^{2+}$ | $Ca^{2+}$ | $Cl^-$ | $SO_4^{2-}$ | $HCO_3^-$ | $CO_3^{2-}$ | TDS | pH |
|---|---|---|---|---|---|---|---|---|---|---|
| Minimum | 10.10 | 0.38 | 0.72 | 2.51 | 4.98 | 6.26 | 182.00 | 0.00 | 285.00 | 7.34 |
| Maximum | 834.00 | 9.73 | 40.00 | 147.00 | 641.00 | 1120.00 | 481.00 | 14.60 | 2985.00 | 8.87 |
| Mean | 76.18 | 1.66 | 15.62 | 55.94 | 40.60 | 82.77 | 256.74 | 1.22 | 551.31 | 7.84 |
| Standard deviation | 138.37 | 1.63 | 8.86 | 32.16 | 98.22 | 192.46 | 59.33 | 3.54 | 439.34 | 0.33 |

The dissolution of carbonate minerals (e.g., calcite and dolomite) releases $Mg^{2+}$, $Ca^{2+}$, and $HCO_3^-$ into the groundwater. The average concentration (15.62 mg/L) of $Mg^{2+}$ was lower than that (55.94 mg/L) of $Ca^{2+}$. The concentrations of both $Mg^{2+}$ and $Ca^{2+}$ were below their respective limits stipulated in the National Drinking Water Standards. These results show that calcite dissolution is a dominant factor in the groundwater environment. On the other hand, gypsum dissolution is another source of $Ca^{2+}$ in the groundwater. The concentration of $SO_4^{2-}$ ranged from 6.26 to 1120 mg/L, averaging 82.77 mg/L (higher than the average concentration of $Ca^{2+}$), indicating possible precipitation or cation-exchange reactions of Ca-bearing minerals or the presence of other sources of $SO_4^{2-}$ (e.g., mirabilite) in the groundwater runoff. $HCO_3^-$ in most natural groundwater bodies originates from the dissolution of the $CO_2$ from the atmosphere and the vadose zone and carbonate minerals [3]. The average concentration of $HCO_3^-$ in the groundwater in the study area was 256.74 mg/L.

The $Na^+$ and $Cl^-$ in meteoric water infiltrating into groundwater were nearly equal in concentration. The groundwater samples collected from most sampling sites were near the 1:1 line in Figure 3a, suggesting that the groundwater originated primarily from meteoric water. In addition, halite dissolution releases $Na^+$ and $Cl^-$ in equal concentration to the groundwater. Groundwater samples collected from some sampling sites were below the 1:1 line, indicating that the hydrolysis of other Na-containing minerals had led to an excess of $Na^+$. The weathering and dissolution of carbonate minerals (e.g., calcite and dolomite) is the principal source of $Ca^{2+}$ and $Mg^{2+}$ in groundwater. The ratio of the combined milligram equivalent concentration of $Ca^{2+}$ and $Mg^{2+}$ to the milligram equivalent concentration of $HCO_3^-$ was found to be 1.

The groundwater samples collected from most of the sampling sites were near the 1:1 line of Figure 3b, while samples retrieved from some sampling sites were near the 1:2 or 2:1 line. That the combined concentration of $Ca^{2+}$ and $Mg^{2+}$ was higher than the concentration of $HCO_3^-$ may be attributed to the dependence of the weathering and dissolution of carbonate minerals or Ca and Mg feldspars on weak acids instead of strong acids or the presence of other sources of $Ca^{2+}$ and $Mg^{2+}$ (e.g., gypsum). That the combined concentration of $Ca^{2+}$ and $Mg^{2+}$ was lower than the concentration of $HCO_3^-$ may be attributed to the increase in the concentration of $HCO_3^-$ caused by silicate dissolution or the decrease in the combined concentration of $Ca^{2+}$ and $Mg^{2+}$ resulting from cation exchange.

The groundwater samples collected from most of the sampling sites were near the 1:1 line of Figure 3c, though those from some sampling sites were above the 1:1 line, suggesting the dissolution of other $SO_4^{2-}$-containing minerals in the groundwater in addition to gypsum. Further, the groundwater samples collected from all the sampling sites were near the 1:1 line in Figure 3d, except for one outlier sample, indicating that mirabilite dissolution was the source of the excess of $Na^+$ in the groundwater at some sampling sites. Under normal circumstances, the concentration of $Cl^-$ is stable in the groundwater environment, and $Cl^-$ does not undergo chemical or physical reactions with other ions or minerals. Groundwater samples collected from some sampling sites were above the 1:1 line, indicating the dissolution of other sulfates. Figure 3e mainly explains the linear relationship between the combined concentration of $Ca^{2+}$ and $Mg^{2+}$ originating from sources other than the dissolution of carbonates and feldspars and the concentration of $SO_4^{2-}$ originating from sources other than mirabilite dissolution. The groundwater samples were near this 1:1 line, suggesting gypsum dissolution in the groundwater.

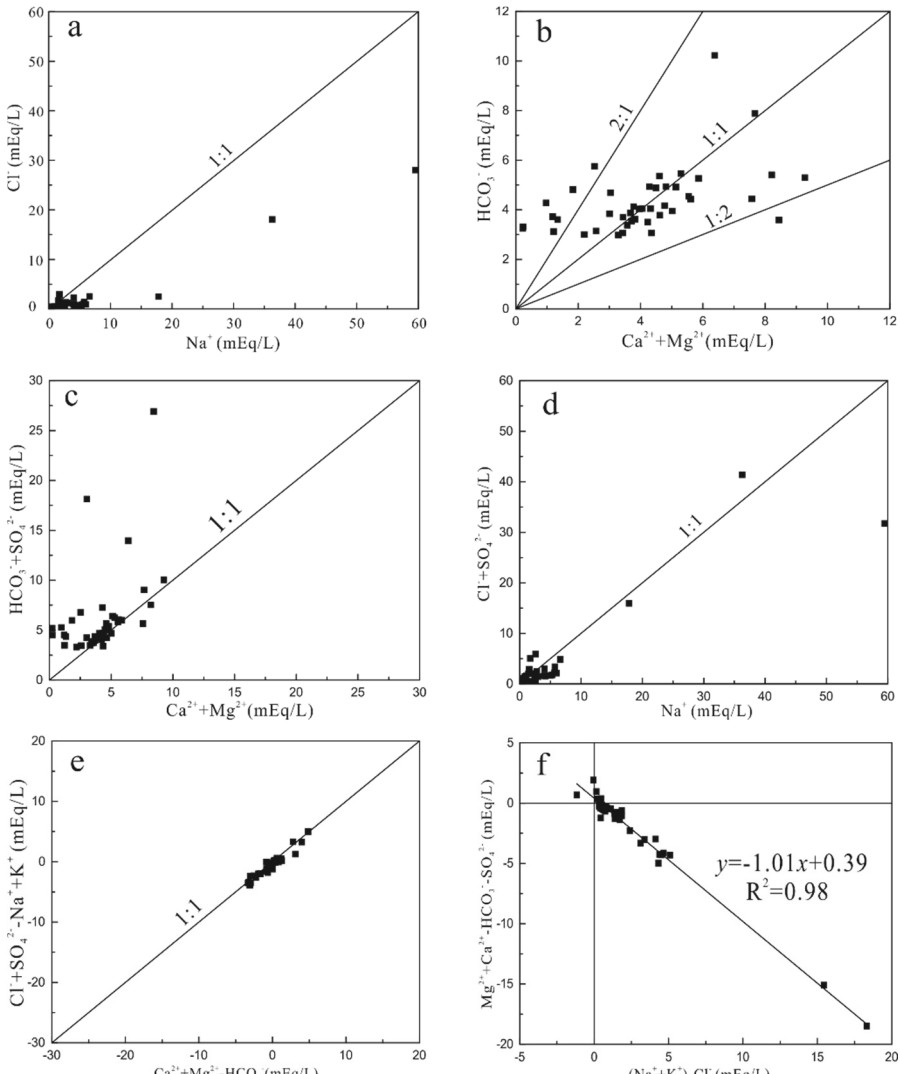

**Figure 3.** Relationships between different ions with *solid lines* representing the theoretical dissolution *curves* ((**a**). $Na^+$ vs. $Cl^-$ (**b**). $Ca^{2+} + Mg^{2+}$ vs. $HCO_3^-$ (**c**). $Ca^{2+} + Mg^{2+}$ vs. $HCO_3^- + SO_4^{2-}$ (**d**). $Na^+$ vs. $Cl^- + SO_4^{2-}$ (**e**). $Ca^{2+} + Mg^{2+} - HCO_3^-$ vs. $Cl^- + SO_4^{2-} - Na^+ + K^+$ (**f**). $(Na^+ + K^+) - Cl^-$ vs. $Ca^{2+} + Mg^{2+} - HCO_3^- - SO_4^{2-}$).

Figure 3f reflects whether cation exchange, specifically between $Na^+$ and $Ca^{2+}$, occurred in the groundwater. A slope near $-1$ indicates the occurrence of cation exchange at a groundwater sampling site. As shown in Figure 3f, the slope of the fitted curve corresponding to each groundwater sampling site was $-1.01$, suggesting an exchange between $Na^+$ and $Ca^{2+}$ during the groundwater runoff process.

### 4.2. Formation of the Hydrochemical Composition of the Groundwater

As seen in Figure 4, groundwater samples collected from most sampling sites fall in the water–rock interaction region on the Gibbs diagrams (TDS vs. $Na^+/(Na^+ + Ca^{2+})$), suggesting that the major constituents of the groundwater in the study area originate primarily from water–rock interactions that occur over a long time after the groundwater is recharged by infiltrated precipitation. Sampling site D69 was located in the upper reaches of the Zhasake River and at the northern boundary of the study area, where the groundwater was shallow. The concentration of total dissolved solids (TDS) at site D69 was 1499 mg/L. Sampling site D81 was located in the upper reaches of the Mudushili River and the heavily eroded northern part of the study area. The intense water–rock interactions and evaporation-induced concentration process in the piedmont recharge zone led to a high

concentration of TDS (2985 mg/L, as found in this study) at site D81. The hydrochemical composition of the groundwater at most sampling sites was controlled by water–rock interactions. The ratio of the concentration of $Na^+$ to the combined concentration of $Na^+$ and $Ca^{2+}$ ($Na^+/(Na^+ + Ca^{2+})$) was above 0.5 at most sampling sites, suggesting an exchange between $Na^+$ and $Ca^{2+}$ in the groundwater runoff [29,30].

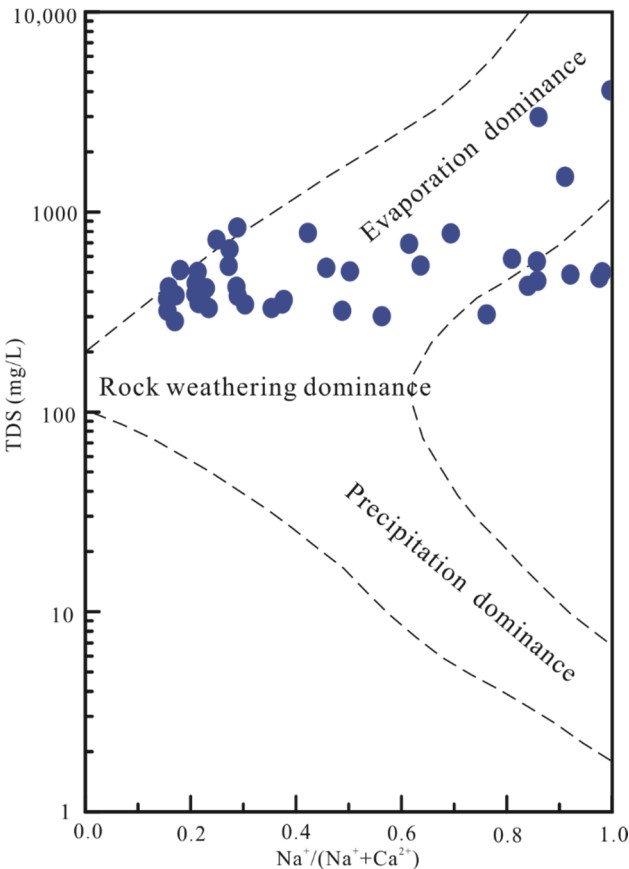

**Figure 4.** Gibbs diagrams for the groundwater in the study area.

Gaillardet et al. [31] proposed using a Na-normalized molar ratio to reflect different hydrochemical reactions under non-mixed conditions. Compared to the Gibbs diagram, which can only indicate whether water–rock interactions are the dominant mechanism overall, this metric can identify particular water–rock interactions. The Gaillardet diagram consists of three regions that correspond to three respective hydrochemical processes: carbonate dissolution, silicate weathering, and evaporite dissolution. If a groundwater sample falls in a certain region of the graph, it means that the hydrochemical process depicted by that region plays a dominant role at the site where the sample was collected. If a groundwater sample falls between two regions, it suggests the coexistence of different hydrochemical processes at the site. As seen in Figure 5, groundwater samples collected from most sampling sites in the study area fell between the silicate weathering and carbonate dissolution regions on the Gaillardet diagram, suggesting that the composition of the groundwater in the study area is controlled predominantly by silicate weathering and carbonate dissolution. Due to their high concentrations of TDS, groundwater samples collected from some sampling sites (i.e., D63, D69, and D81) fell near the evaporite dissolution region.

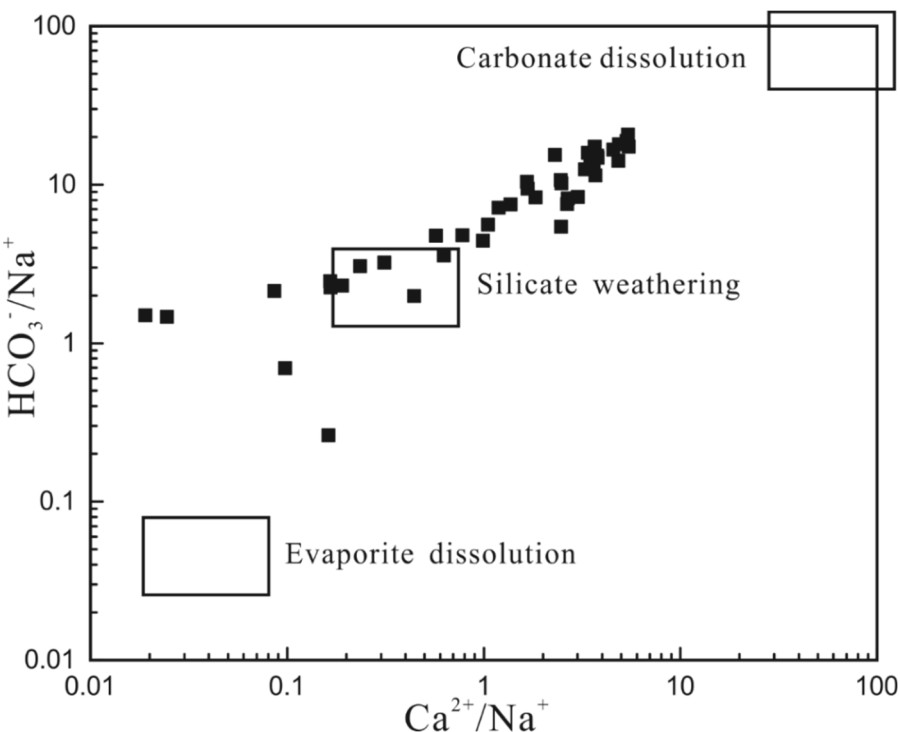

**Figure 5.** Gaillardet diagram for the groundwater in the study area.

Cation exchange is a process in which particles adsorb some cations from the groundwater and simultaneously release some of the previously adsorbed cations back into the groundwater under certain conditions. The CAIs (CAI 1 and CAI 2) introduced by Schoeller [32] (see Equations (2) and (3)) were used to further investigate the cation exchange in the groundwater in the study area:

$$CAI1 = \frac{Cl^- - (Na^+ + K^+)}{Cl^-} \tag{2}$$

$$CAI2 = \frac{Cl^- - (Na^+ + K^+)}{HCO_3^- + SO_4^{2-} + CO_3^{2-} + NO_3^-} \tag{3}$$

If both CAI 1 and CAI 2 are positive, it means that there is an exchange between the $Na^+$ and $K^+$ in the groundwater and the $Ca^{2+}$ and $Mg^{2+}$ in the surrounding rock; otherwise, it suggests an exchange between the $Ca^{2+}$ and $Mg^{2+}$ in the groundwater and the $Na^+$ and $K^+$ in the surrounding rock. High absolute values of CAI 1 and CAI 2 indicate a high cation exchange tendency [10].

As seen in Figure 6, the CAI 1 and CAI 2 values for groundwater samples collected from most sampling sites were below 0, suggesting that the cation exchange in the groundwater environment in the study area was dominated by the exchange between the $Ca^{2+}$ and $Mg^{2+}$ in the groundwater and the $Na^+$ and $K^+$ in the surrounding rocks. This result also pinpoints the source of the excess $Na^+$ in the groundwater. The CAI 1 and CAI 2 values for the samples collected at sites D25, D48, and D49 were positive, suggesting an opposite exchange.

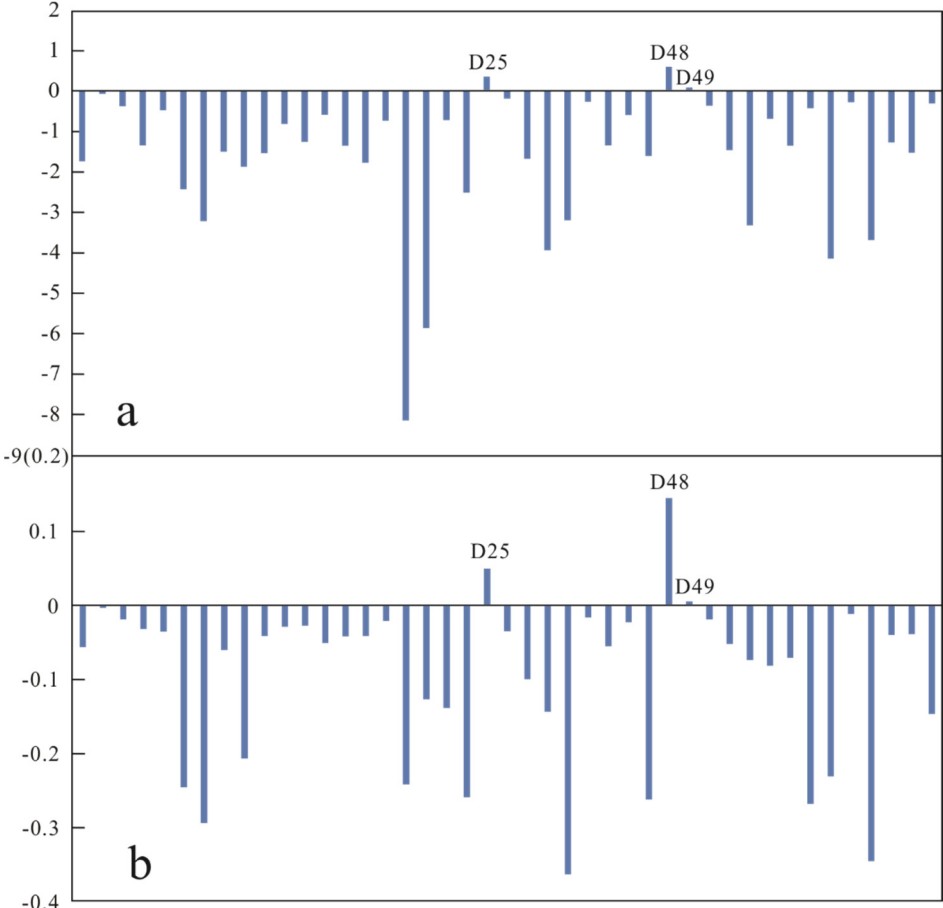

**Figure 6.** Cation exchange diagrams for the groundwater in the study area ((**a**). The value of CAI 1 (**b**). The value of CAI 2).

### 4.3. Analysis of Mineral Dissolution

The chemical reactions between groundwater and its surrounding rocks can be used to elucidate the characteristics of their interactions and to reveal the pattern of evolution of the hydrochemical composition of the groundwater. The hydrogeochemical simulation software PHREEQC can be used to simulate the geochemical processes in a groundwater system. The hydrogeochemical code PHREEQC is employed to conduct the calculation of SI speciation by using the database of phreeqc.dat. It calculates the SI for a given mineral under different controlled conditions based on hydrochemical data (Appendix A, Table A1) for the groundwater to reflect the equilibrium state of the mineral. On this basis, the role of one or multiple reactive minerals in controlling the hydrochemical composition of the groundwater can be determined [3,19]. The *SI* for a mineral can be calculated using Equation (4) to determine whether it dissolves or precipitates in the groundwater:

$$SI = \lg \frac{IAP}{K} \tag{4}$$

where *IAP* is the ion activity product and *K* is the equilibrium constant. A positive *SI* indicates that the mineral is oversaturated in the groundwater, tends to precipitate from it, and maybe is nonreactive; a negative *SI* suggests that the mineral is unsaturated in the groundwater and will thus tend to dissolve. If the absolute value of the *SI* falls within 0–0.5, the mineral is considered to be in an equilibrium state. A high absolute value of the *SI* suggests significant dissolution or precipitation of the mineral [3].

Table 2 summarizes the calculated values of the SI for gypsum, calcite, dolomite, halite, and fluorite. The values of the SI for halite, gypsum, and fluorite were all negative, while the values of the SI for calcite and dolomite ranged from negative to positive. The values

of the SI for evaporite halite ranged from $-8.86$ to $-4.55$, averaging $-7.59$, suggesting considerable halite dissolution in the groundwater of the study area, which explains its high concentrations of $Na^+$ and $Cl^-$. The values of the SI for gypsum and fluorite ranged from $-3.14$ to $-0.74$ and from $-4.12$ to $-1.10$, respectively, with averages of $-2.25$ and $-2.29$, indicating that these two minerals also actively dissolve in the groundwater and are the main sources of $Ca^{2+}$. The underground water-bearing media in the study area are rich in halite, gypsum, and fluorite. These minerals are the major sources of ions in the groundwater. These conclusions are consistent with the results of the earlier hydrochemical analysis of the groundwater.

**Table 2.** Calculated SI values for selected minerals.

|         | Calcite | Halite | Gypsum | Fluorite | Dolomite |
|---------|---------|--------|--------|----------|----------|
| Minimum | $-0.24$ | $-8.86$ | $-3.14$ | $-4.12$ | $-0.62$ |
| Maximum | $1.09$ | $-4.55$ | $-0.74$ | $-1.10$ | $2.13$ |
| Mean | $0.44$ | $-7.59$ | $-2.25$ | $-2.29$ | $0.70$ |

To further illustrate the relationships between the mineral dissolution or precipitation and the ions in the groundwater, the SI for each mineral and the concentrations of the corresponding ions were plotted in the same graph (Figure 7). As seen in Figure 7a, the SI for halite was positively correlated with the combined concentration of $Na^+$ and $Cl^-$, suggesting that halite dissolution is the primary source of $Na^+$ and $Cl^-$ in the groundwater. As the combined concentration of $Na^+$ and $Cl^-$ in the groundwater increased, the SI for halite increased first sharply and then more slowly after a certain total concentration of $Na^+$ and $Cl^-$ was reached. A similar linear relationship was found between the SI for gypsum and the combined concentration of $Ca^{2+}$ and $SO_4^{2-}$ (Figure 7b), indicating that gypsum actively dissolves in the groundwater and is the primary source of $Ca^{2+}$ and $SO_4^{2-}$. Fluorite dissolution can also provide $Ca^{2+}$ to groundwater. Figure 7c reveals a positive correlation between the SI for fluorite and the combined concentration of $Ca^{2+}$ and $F^-$. The groundwater data points in Figure 7c are scattered, unlike those in Figure 7a,b, which are concentrated and exhibit strong linear relationships, indicating that compared to halite and gypsum, fluorite is the predominant source of ions in the groundwater.

The values of the SI for the carbonate minerals—calcite and dolomite—were mostly positive (Figure 7), suggesting that these minerals may exist in the water-bearing media and control the hydrochemical composition of the groundwater. As seen in Figure 7d, the SI for calcite in the groundwater at three sampling sites was negative and gradually increased with an increase in the total concentration of $Ca^{2+}$ and $HCO_3^-$, indicating that most calcite in the groundwater exists in the form of precipitates or is reactive. The SI for dolomite in the groundwater at six sampling sites was negative (Figure 7e). The variation in dolomite with the change in the total concentration of the corresponding ions is similar to that of calcite with the total concentration of $Ca^{2+}$ and $HCO_3^-$, suggesting that dolomite also precipitates in the groundwater. The carbonate minerals in the groundwater in the study area were nonreactive or precipitates. In addition, the groundwater environment had a low combined concentration of $Ca^{2+}$ and $Mg^{2+}$ and a high concentration of $HCO_3^-$. These findings are consistent with the earlier analysis of the hydrochemical characteristics of the groundwater and the formation of its hydrochemical composition. Silicate minerals are dissolved in the groundwater. Moreover, hydrogeological drilling data reveal that the sandstones in the Cretaceous Luohe and Huanhe formations contain feldspar minerals. The precipitation of calcite and dolomite decreases the combined concentration of $Ca^{2+}$ and $Mg^{2+}$ in the groundwater and yields a large amount of free $CO_2$. Excessive $CO_2$ dissolution in the groundwater promotes the dissolution of Na and K feldspars.

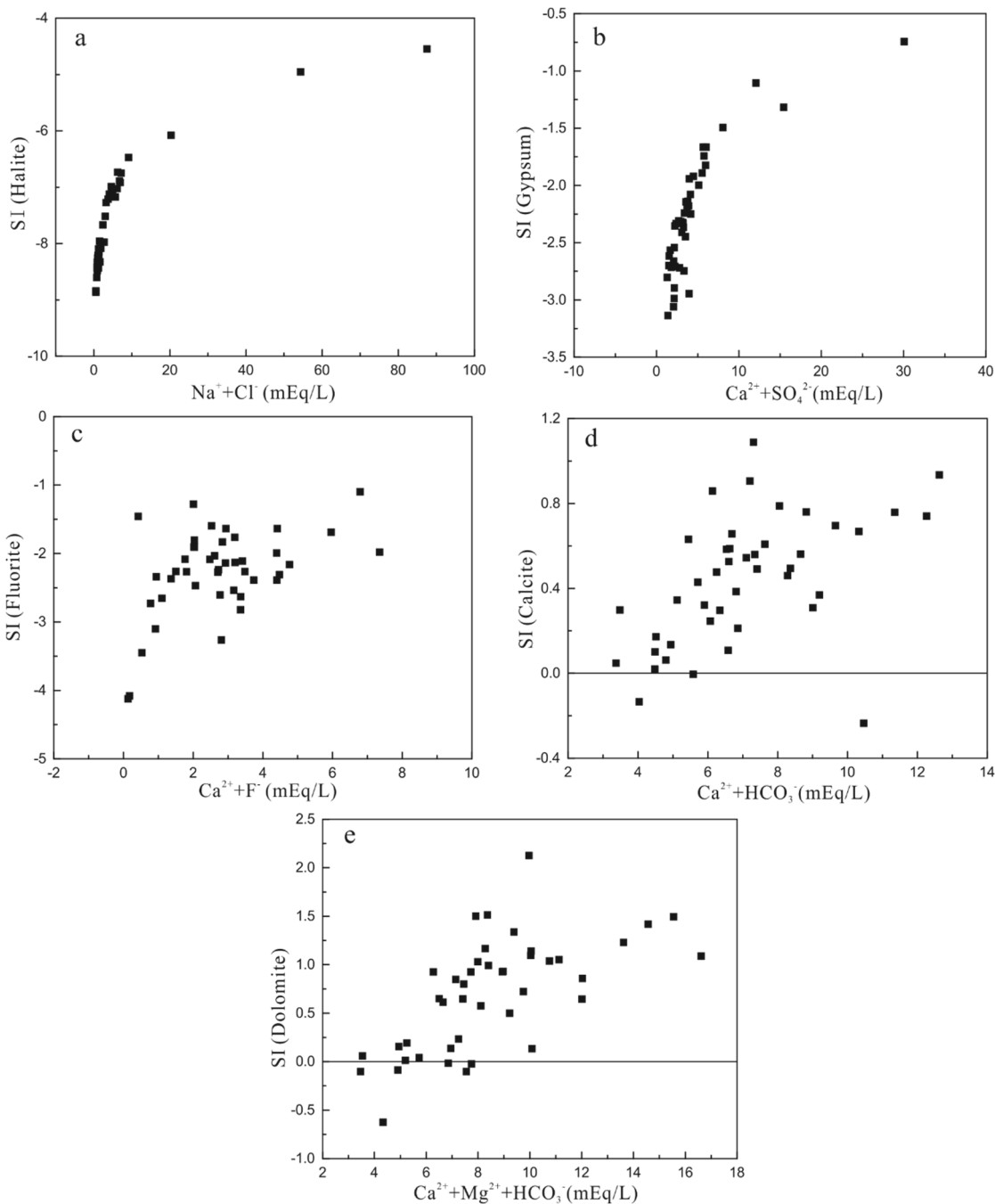

**Figure 7.** SI plots for the groundwater in the study area ((**a**). Na$^+$ + Cl$^-$ vs. SI (Halite) (**b**). Ca$^{2+}$ + SO$_4^{2-}$ vs. SI (Gypsum) (**c**). Ca$^{2+}$ + F$^-$ vs. SI (Fluorite) (**d**). Ca$^{2+}$ + HCO$_3^-$ vs. SI (Calcite) (**e**). Ca$^{2+}$ + Mg$^{2+}$ + HCO$_3^-$ vs. SI (Dolomite)).

### 4.4. Hydrochemical Facies of Groundwater

Groundwater samples collected from most sampling sites fell in region 1 on the Piper trilinear diagram in Figure 8, corresponding to an HCO$_3$–Ca facies. The samples collected from sites D69 and D81 fell in region 2, corresponding to a Cl–Na facies, while those collected from sites D10, D21, and D39 fell in region 3, corresponding to an HCO$_3$–Na facies. The other water samples fell in region 5, corresponding to mixed hydrochemical facies. Calculations performed based on an ion milligram equivalent percentage of over 25% revealed an HCO$_3$–SO$_4$–Ca facies at site D07, an HCO$_3$–Cl–Na–Ca facies at site D23, an HCO$_3$–SO$_4$–Na facies at site D47, an HCO$_3$–Cl–Na facies at site D63, and an HCO$_3$–Na–Ca facies at sites D09, D13, D24, and D66. In addition, the anion and cation trilinear diagrams

similarly showed that Ca, Na, and Ca–Na were the dominant cations and that HCO$_3$ was the dominant anion.

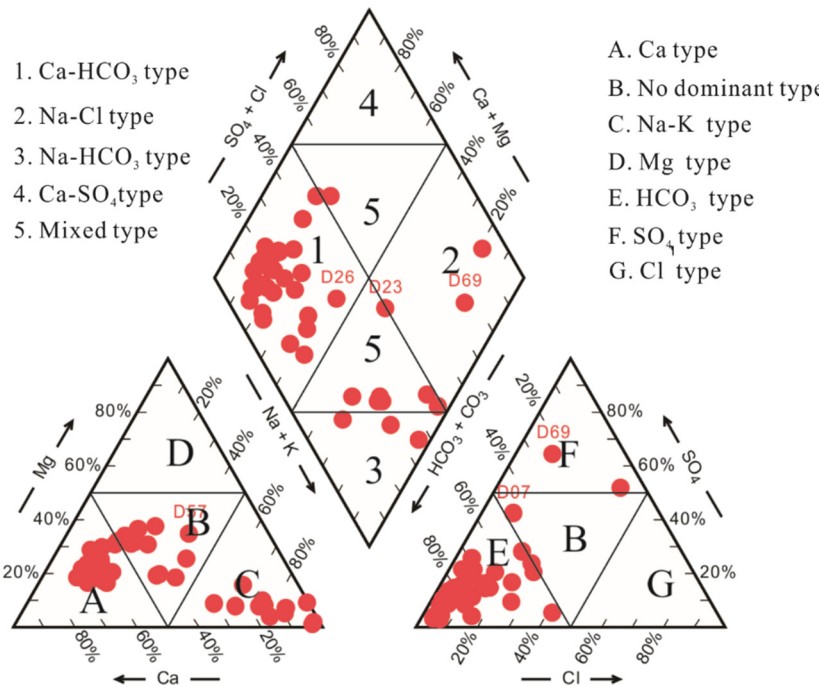

**Figure 8.** Piper diagram for the groundwater in the study area.

The spatial variation of the hydrochemical facies of the groundwater was investigated based on the above hydrochemical facies analysis combined with the groundwater contour map. Groundwater flows from site D10, located on the shore of the Hongjiannao Lake, to Hongjiannao Lake and the nearby lake water sampling site D34. The hydrochemical facies of the groundwater transition from HCO$_3$–Na at site D10 to Cl–Na at site D34. Analysis of the SI for the minerals at site D34 revealed a positive SI value for each of calcite and dolomite and a negative SI value for halite, suggesting that the HCO$_3^-$ in the groundwater runoff combines with Ca and Mg to form precipitates and that Cl$^-$ replaces HCO$_3^-$ and combines with Na$^+$ to form a new hydrochemical facies. HCO$_3$–Ca was the hydrochemical facies at both sampling sites D22 and D26, located in the recharge zone for the groundwater at site D23, which is of the HCO$_3$–Cl–Na–Ca mixed facies. Analysis showed positive SI values for calcite and dolomite and negative SI values for halite at these three sampling sites, suggesting that halite actively dissolves in the groundwater runoff, releasing more Na$^+$ and Cl$^-$, which in turn leads to a transition in the hydrochemical facies to a mixed combination of HCO$_3$–Ca and Cl–Na. The above analysis based on the SI values of minerals reveals a spatial variability in the hydrochemical facies of the groundwater in the study area.

### 4.5. SOM-Based Clustering

SOMs were trained on the hydrochemical data for the groundwater sampled in the rainy and dry seasons and were then used to normalize these data. SOMs for eight hydrochemical parameters were obtained (Figure 9). In each map, the color shade of each neuron represents the component value of the hydrochemical parameter of the groundwater at the sampling site. These maps visually display the distances between the corresponding neurons and the distribution of their color shades and elucidate the information and qualitative relationships between the hydrochemical parameters. In Figure 9, the SOMs for the concentrations of Ca$^{2+}$, Mg$^{2+}$, K$^+$, and HCO$_3^-$ display similar color gradients, suggesting strong correlations between the concentrations of these ions. The correlations between the concentrations of Ca$^{2+}$, Mg$^{2+}$, and HCO$_3^-$ in the groundwater indicate that they may

originate from the dissolution of calcite and dolomite. The correlation between the concentrations of $K^+$ and $HCO_3^-$ suggests the dissolution of K feldspars in the groundwater. On the other hand, the SOM for the concentration of $CO_3^{2-}$ exhibits a color gradient opposite to those of the SOMs for the concentrations of the other seven ions, indicating that the concentration of $CO_3^{2-}$ is negatively correlated with each of the other seven ion concentrations. The SOMs for the $Na^+$, $Cl^-$, and $SO_4^{2-}$ concentrations displayed similar color gradients, suggesting strong positive correlations between them. The positive correlation between the concentrations of $Na^+$ and $Cl^-$ was primarily a result of halite dissolution in the groundwater, while the correlation between the $Na^+$ and $SO_4^{2-}$ concentrations may be attributed to mirabilite dissolution. $Ca^{2+}$ and $SO_4^{2-}$ originate from gypsum dissolution. The SOMs revealed a weak positive correlation between the concentrations of these two ions.

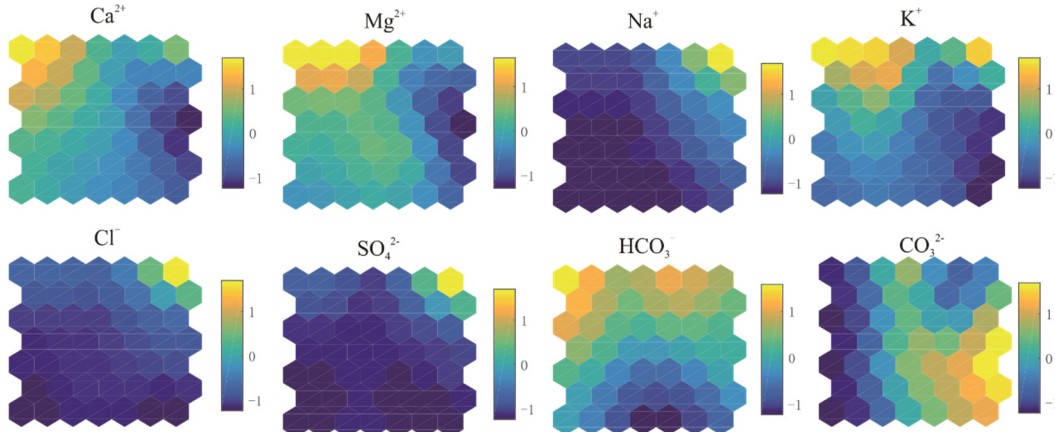

**Figure 9.** SOMs for the groundwater in the study area.

Meanwhile, the correlation heatmap was plotted using the hydrochemical data, as shown in Figure 10. The correlation heatmap of the same variable is shown by the largest and darkest solid circle. As the correlation decreases, the solid circles become smaller and lighter in color, so it can intuitively check the correlation between different variables. There was a strong positive correlation between the concentrations of any two of $Ca^{2+}$, $Mg^{2+}$, $K^+$, $HCO_3^-$, $Cl^-$, and $SO_4^{2-}$. The concentration of $CO_3^{2-}$ was negatively correlated with that of each of the other ions besides $HCO_3^-$. The concentration of $Na^+$ was most strongly positively correlated with the concentration of each of $HCO_3^-$, $Cl^-$, and $SO_4^{2-}$. The results of the correlation heatmap are basically consistent with those of the SOM-based analysis, suggesting that the SOMs can be used to represent the correlations between these parameters.

The trained SOM-normalized hydrochemical data for the groundwater in the study area in the rainy and dry seasons were used as the input matrix X in the calculation of the DBI. We set $N$ equal to 10. Figure 11 shows the calculated values of the DBI. As seen in Figure 11a, the minimum DBI occurred at $N_c = 5$. Therefore, $N_c$ was set to 5 for the cluster analysis in this study. Based on the optimal $N_c$, the groundwater samples collected during the rainy and dry seasons were clustered through SOM-based calculations. In Figure 11b, the five clusters are distinguished with different colors. The neurons covered with the same color represent one cluster. Each cluster contains groundwater samples with similar hydrogeochemical characteristics.

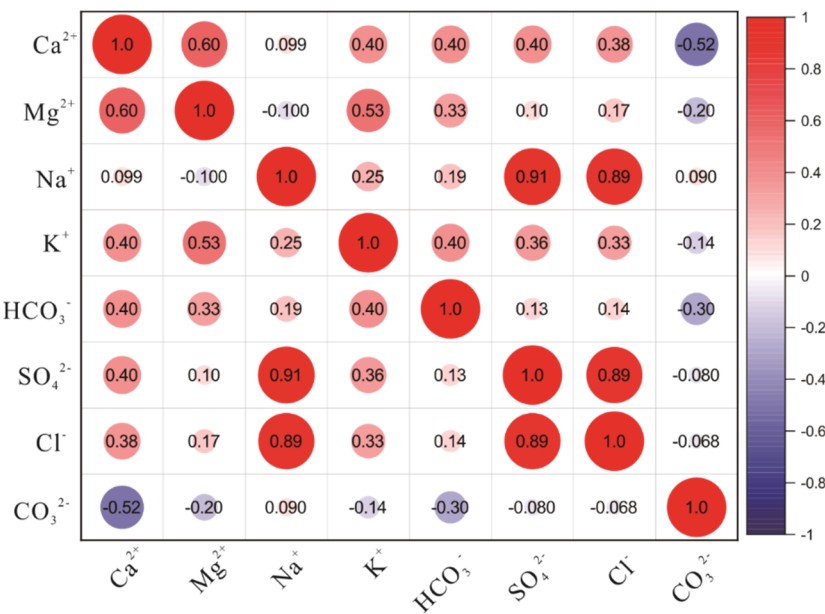

**Figure 10.** Correlation heatmap of hydrochemical parameters.

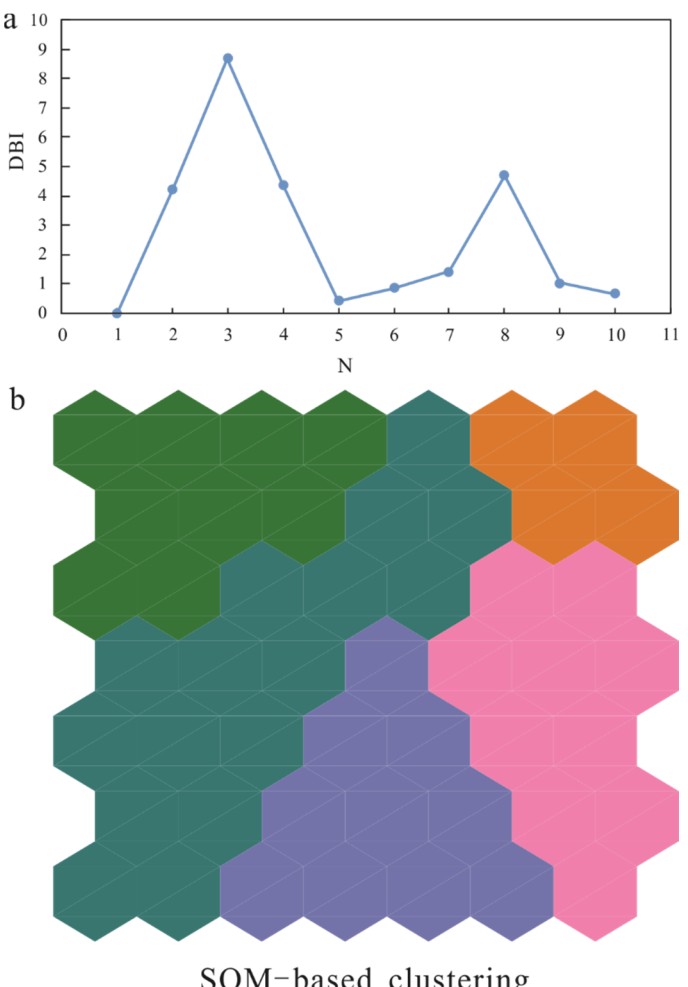

**Figure 11.** SOM-based cluster diagram ((**a**). The value of the DBI (**b**). Different colors indicate different clusters).

Figure 12 shows the SOMs for the groundwater samples collected from the study area during the rainy and dry seasons. This diagram was produced through repetitive iterations, self-organizing learning, and training based on the SOM algorithms to facilitate the visual output and cluster analysis. Groundwater samples with similar hydrogeochemical characteristics were assigned to the same SOM neuron (the suffixes R and D signify rainy- and dry-season samples, respectively). The temporal and spatial variability of the hydrochemical composition of the groundwater in the study area was further analyzed based on the distance between the neurons containing the samples collected during different seasons at each sampling site. Figure 12b gives a visual representation of the clustering of the groundwater samples collected from the study area during the rainy and dry seasons based on a combination of Figures 11b and 12a. It shows that the groundwater samples were grouped into five clusters as well as which groundwater samples made up each cluster. Analysis based on the locations of the groundwater samples depicted in the SOMs in Figure 9 identified $Ca^{2+}$, $Mg^{2+}$, $K^+$, and $HCO_3^-$ as the dominant ions in the groundwater samples in cluster 1 and $Na^+$, $K^+$, $Cl^-$, and $SO_4^{2-}$ as the dominant ions in the groundwater samples in cluster 5.

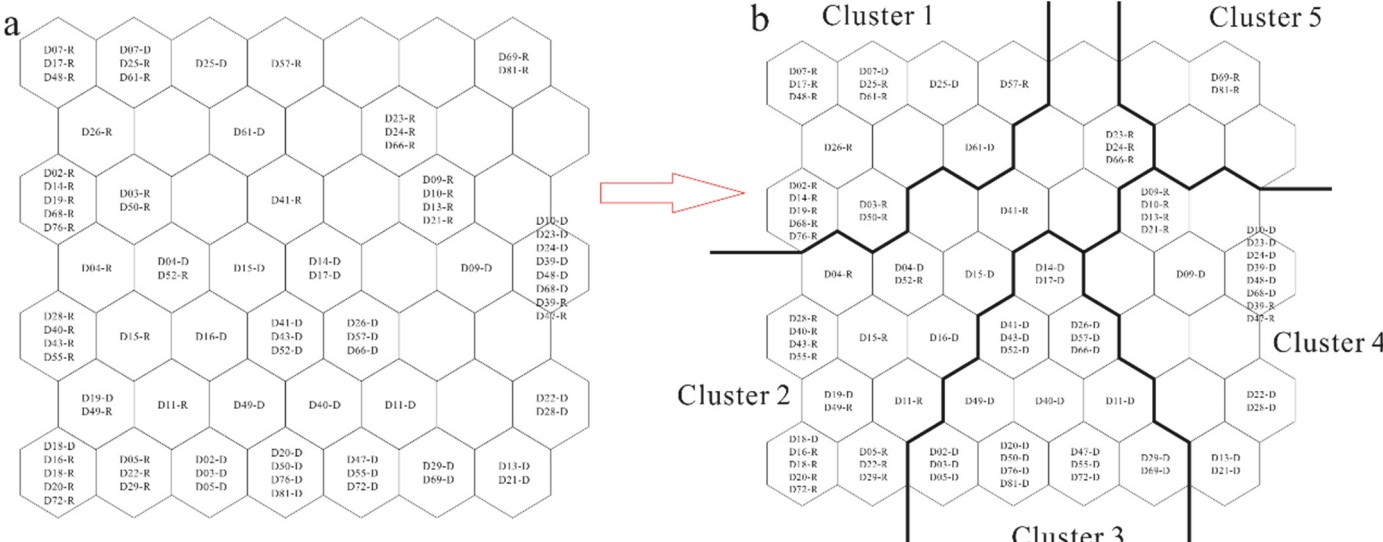

**Figure 12.** SOM-based clustering of the groundwater samples collected during the rainy and dry seasons ((**a**). Different neurons contain samples (**b**). Different clusters contain samples).

### 4.6. Analysis of Clustered Hydrochemical Characteristics

To comprehensively evaluate the hydrochemical clustering of the groundwater in the study area in the rainy and dry seasons, a Piper trilinear diagram was produced based on the hydrochemical data for each SOM-yielded cluster of groundwater samples (Figure 13). The groundwater samples in cluster 1 mostly fell in region 1, with their anions falling in region E and their cations falling in regions A and B, suggesting $HCO_3$–Ca as their hydrochemical facies. This finding is consistent with the earlier analysis of the groundwater samples in this cluster: $Ca^{2+}$, $Mg^{2+}$, and $HCO_3^-$ were identified as their dominant ions. Similarly, $HCO_3$–Ca was the hydrochemical facies of the groundwater samples in cluster 2. The Piper diagram reveals the presence of weak cations in the groundwater samples in cluster 2, which is consistent with the conclusion drawn from the color gradient in the SOM for the concentration of $Ca^{2+}$ in Figure 9. The groundwater samples in cluster 3 fell in the same region as those in cluster 1, suggesting $HCO_3$–Ca as their hydrochemical facies. The groundwater samples in cluster 4 mostly fell on the border between regions 4 and 5, their cations falling in region D and their anions falling in region E, suggesting $HCO_3$–Na as their hydrochemical facies, which is consistent with the SOMs for the concentration of $Na^+$ and $HCO_3^-$. Cluster 5 contained only two groundwater samples. They fell in

region 3, their cations falling in region D and their anions falling in region F, indicating Cl–Na as their hydrochemical facies. Overall, $HCO_3$–Ca type and $HCO_3$–Na type were the dominant hydrochemical facies of the groundwater in the study area during the rainy and dry seasons.

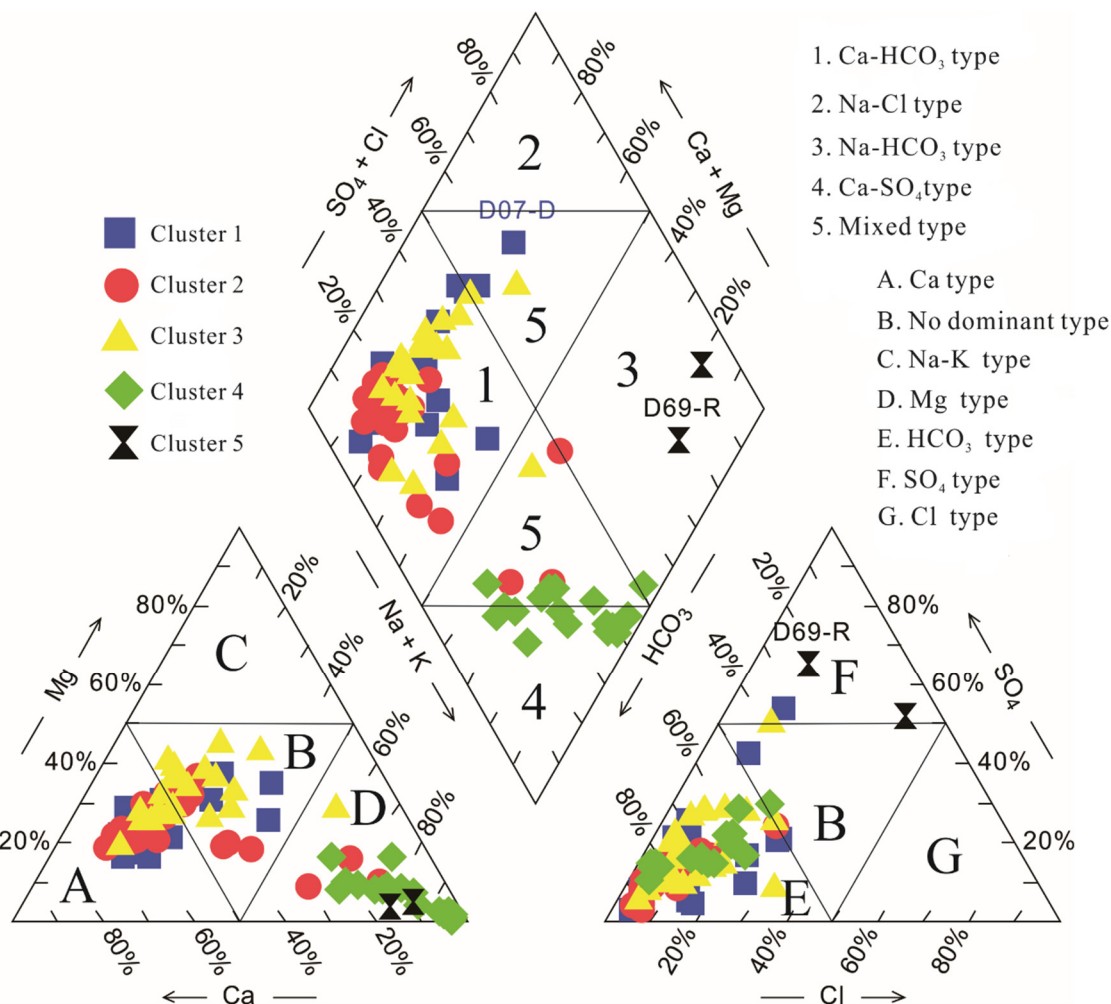

**Figure 13.** Piper diagram for the groundwater in the rainy and dry seasons.

Figure 14 shows a Gibbs diagram for the clusters obtained based on the SOMs (TDS vs. $Na^+/(Na^+ + Ca^{2+})$). The groundwater samples in clusters 1, 2, 3, and 4 mostly fell in the region corresponding to water–rock interaction dominance. The groundwater samples in cluster 1 fell on the edge of the evaporation-induced concentration region, suggesting that the groundwater at the sites where these samples were collected is also affected by the evaporation-induced concentration process. Of the groundwater samples in clusters 1, 2, and 3, those in cluster 1 had the highest average concentration of TDS, followed by those in clusters 2 and 3. Based on the SOMs (in the upper-left corner of Figure 9), the high concentration of TDS in the groundwater samples in cluster 1 can be attributed to their high concentrations of $Ca^{2+}$, $Mg^{2+}$, and $HCO_3^-$. With a $Na^+/(Na^+ + Ca^{2+})$ ratio of greater than 0.7, the groundwater samples in cluster 4 fell outside the region encircled by the dotted lines, suggesting intense cation exchange in the groundwater environment where these samples were collected. With a high $Na^+/(Na^+ + Ca^{2+})$ ratio, the groundwater samples in cluster 5 fell in the evaporation-induced concentration region, suggesting that the hydrochemical composition of the groundwater where these two samples were collected is dually affected by the evaporation-induced concentration process and cation

exchange. Overall, the hydrochemical composition of the groundwater in the study area during the rainy and dry seasons is primarily controlled by water–rock interactions.

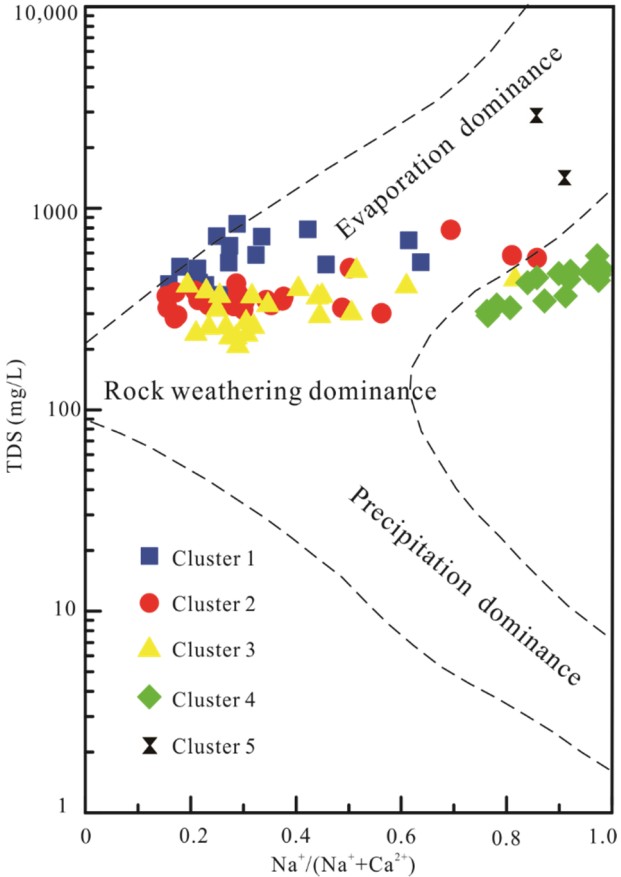

**Figure 14.** Gibbs diagram for the groundwater during the rainy and dry seasons.

*4.7. Seasonal Variability*

The seasonal variability of the groundwater in the study area during the rainy and dry seasons was further analyzed. Figure 15 visually depicts the changes in the clustering of the sampling sites corresponding to the change from the rainy season to the dry season, which was determined based on the topological distances between the SOM neurons in combination with the cluster analysis. Changes can be observed in the clustering of 30 sampling sites. The blue arrows show the changes in the clustering of the sampling sites in cluster 1 (a total of 11 sampling sites) corresponding to a change from the rainy season to the dry season. A change from cluster 1 to cluster 2 was observed in the assignment of one sampling site (D19), while a change from cluster 1 to cluster 3 was observed in the assignment of eight sampling sites. Based on the above analysis of the hydrochemical facies of the groundwater and the formation of its hydrochemical composition, there was weak seasonal variability in the hydrochemical characteristics of the groundwater at these nine sampling sites. A change from cluster 1 to cluster 4 was observed in the assignment of two sampling sites, corresponding to a change in the hydrochemical facies from $HCO_3$–Ca to $HCO_3$–Na, suggesting that the formation of the hydrochemical composition of the groundwater at these sites is accompanied by an intense cation-exchange process and that there is a strong seasonal variability in the hydrochemical characteristics of the groundwater at these two sites.

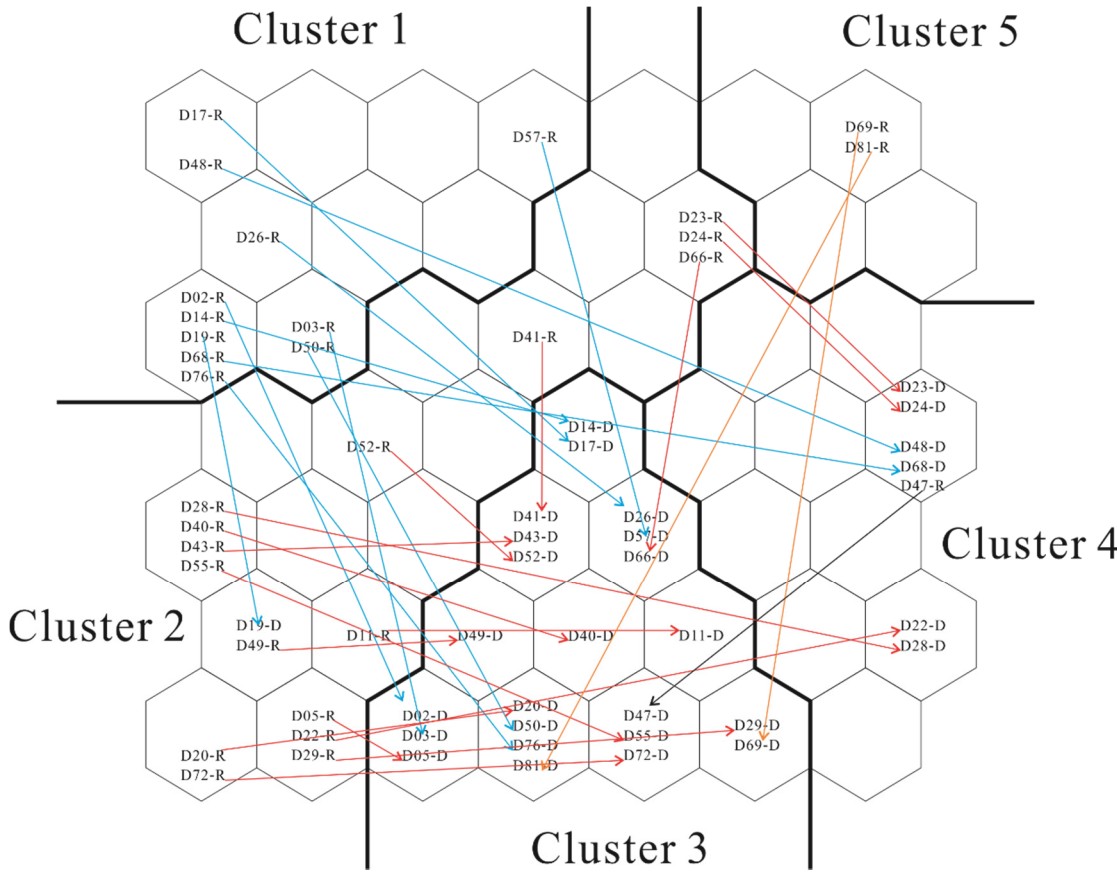

**Figure 15.** Variations in the groundwater in the study area with the season.

The red arrows in Figure 15 show the changes in the clustering of the sampling sites in cluster 2 (a total of 11 sampling sites) corresponding to a change from the rainy season to the dry season. A change from cluster 2 to cluster 3 was observed in the assignment of 12 sampling sites, suggesting no significant changes in the hydrochemical characteristics of the groundwater at these sites. A change from cluster 2 to cluster 4 was observed in the assignment of four sampling sites, corresponding to a change in the hydrochemical facies from $HCO_3$–Ca type to $HCO_3$–Na type, which suggests that the formation of the hydrochemical composition of the groundwater at these sites is controlled by cation exchange. A change from cluster 5 to cluster 3 was found in the assignment of two sampling sites, corresponding to a change in the hydrochemical facies from Cl–Na to $HCO_3$–Ca, which indicates that the formation of the hydrochemical composition of the groundwater at these sites changed from an evaporation-induced concentration-dominated mechanism to a water–rock-interaction-dominated mechanism and that there is a strong seasonal variability in the hydrochemical characteristics of the groundwater at these sites. A change from cluster 4 to cluster 3 was observed in the assignment of one sampling site, corresponding to a change in the hydrochemical facies from $HCO_3$–Na to $HCO_3$–Ca, which suggests a weak cation-exchange process and a notable seasonal variability in the hydrochemical characteristics of the groundwater at this site. Overall, the SOM-based seasonal variability diagram reveals changes in the clustering of 30 sampling sites and a notable seasonal variability in the hydrochemical characteristics of the groundwater at nine sampling sites. The hydrochemical characteristics of the groundwater in the study area exhibited no significant seasonal variability.

## 5. Conclusions

(1) The formation of the hydrochemical composition of the groundwater in the study area during the rainy and dry seasons is controlled by water–rock interactions and

cation exchange. Three hydrochemical facies, $HCO_3$–Ca type, $HCO_3$–Na type, and Cl–Na type, dominate the groundwater in the study area, whose composition is controlled primarily by silicate weathering and carbonate dissolution. Halite, gypsum, and fluorite are the dominant sources of ions in the groundwater in the study area. Dolomite and calcite exist mostly in the form of precipitates or reactive minerals in the groundwater of the study area, in which a small amount of feldspar is dissolved.

(2) SOMs were used to cluster the data of the hydrochemical parameters of the groundwater in the rainy and wet seasons. Based on the QE, the TE, and the empirical equation, the number of neurons was optimized to $7 \times 7$. The results derived from the neuron matrices are consistent with those of the Pearson correlation analysis. The number of clusters was optimized through DBI minimization. The groundwater samples collected from the study area during the rainy and dry seasons are grouped into five clusters, with $Ca^{2+}$, $Mg^{2+}$, $K^+$, and $HCO_3^-$ identified as the dominant ions in cluster 1 and $Na^+$, $K^+$, $Cl^-$, and $SO_4^{2-}$ identified as the dominant ions in cluster 5.

(3) $HCO_3$–Ca type is the hydrochemical facies of the groundwater samples in clusters 1, 2, and 3, while $HCO_3$–Na type and Cl–Na type are the hydrochemical facies of the groundwater samples in clusters 4 and 5, respectively. Cation exchange is the dominant factor controlling the formation of the hydrochemical composition of the groundwater at the sites where the groundwater samples in cluster 4 were collected, compared to water–rock interactions for the sites where the groundwater samples in other clusters were collected. The clustering of 30 sampling sites changes with the transition from the rainy season to the dry season. Of these sites, significant seasonal variability was observed in the hydrochemical characteristics of the groundwater at nine sites. Overall, there was no significant seasonal variability in the hydrochemical characteristics of the groundwater in the study area.

**Author Contributions:** Conceptualization, C.W. and X.W.; data curation, C.W., X.W. and C.L.; formal analysis, C.W., Q.S., X.H. and L.Y.; funding acquisition, C.W., X.W. and Q.S.; investigation, C.W. and X.W.; methodology, C.W. and X.W.; project administration, C.W. and X.W.; resources, C.W.; software, C.W. and T.Q.; supervision, C.W. and X.W.; validation, C.W.; visualization, C.W. and X.W.; writing—original draft, C.W.; writing—review & editing, C.W. and X.W. All authors have read and agreed to the published version of the manuscript.

**Funding:** This research was supported by the National Key R&D Program of China (No.2017YF100408), Applied Technology Research and Development Program of Heilongjiang Province (GA19C005), National Natural Science Foundation of China (41572227).

**Institutional Review Board Statement:** Not applicable.

**Informed Consent Statement:** Not applicable.

**Data Availability Statement:** Not applicable.

**Conflicts of Interest:** The authors declare no conflict of interest.

## Appendix A

**Table A1.** Statistical summary for the hydrochemical parameters of groundwater samples (sample locations shown in Figure 1; D02-R represents the sample that was taken during the rainy season; and D02-D represents the sample that was taken during the dry season; unit: mg/L).

| Sample | $Ca^{2+}$ | $Mg^{2+}$ | $Na^+$ | $K^+$ | $HCO_3^-$ | $SO_4^{2-}$ | $Cl^-$ | $CO_3^{2-}$ |
|---|---|---|---|---|---|---|---|---|
| D02-R | 42.80 | 15.60 | 33.80 | 1.10 | 93.50 | 94.80 | 16.70 | 0.00 |
| D03-R | 35.00 | 16.80 | 15.40 | 2.04 | 129.00 | 48.40 | 15.70 | 0.00 |
| D04-R | 44.20 | 16.50 | 23.10 | 2.50 | 203.00 | 40.50 | 20.70 | 0.00 |
| D05-R | 37.40 | 10.70 | 9.94 | 1.37 | 142.00 | 29.30 | 5.10 | 0.00 |
| D07-R | 116.00 | 27.40 | 58.40 | 1.70 | 204.00 | 255.00 | 43.30 | 0.00 |
| D09-R | 10.70 | 2.89 | 121.00 | 1.01 | 197.00 | 48.50 | 35.20 | 6.85 |
| D10-R | 13.50 | 6.67 | 120.00 | 1.15 | 244.00 | 42.20 | 27.60 | 14.10 |
| D11-R | 30.90 | 14.10 | 31.60 | 1.11 | 156.00 | 27.80 | 9.11 | 8.98 |
| D13-R | 16.80 | 7.62 | 54.40 | 0.56 | 177.00 | 22.60 | 4.67 | 8.98 |
| D14-R | 76.70 | 13.30 | 18.50 | 0.59 | 220.00 | 51.90 | 8.30 | 10.00 |
| D15-R | 46.60 | 16.00 | 18.30 | 2.09 | 192.00 | 20.20 | 8.17 | 8.08 |
| D16-R | 50.10 | 10.70 | 10.60 | 1.20 | 188.00 | 21.20 | 5.28 | 5.39 |
| D17-R | 62.20 | 16.20 | 18.60 | 1.06 | 209.00 | 45.40 | 10.90 | 8.78 |
| D18-R | 50.90 | 12.30 | 16.90 | 0.85 | 205.00 | 23.20 | 6.97 | 0.00 |
| D19-R | 45.30 | 13.60 | 19.40 | 1.09 | 201.00 | 23.70 | 8.43 | 0.00 |
| D20-R | 41.70 | 11.00 | 12.80 | 0.90 | 109.00 | 36.30 | 7.94 | 4.64 |
| D21-R | 16.20 | 4.65 | 67.50 | 0.41 | 181.00 | 24.80 | 7.14 | 7.46 |
| D22-R | 18.70 | 5.14 | 66.80 | 0.64 | 182.00 | 25.20 | 4.10 | 10.70 |
| D23-R | 5.38 | 1.30 | 139.00 | 0.61 | 221.00 | 54.20 | 37.30 | 11.00 |
| D24-R | 5.12 | 0.05 | 177.00 | 0.67 | 237.00 | 94.50 | 37.90 | 13.30 |
| D25-R | 79.00 | 35.30 | 37.80 | 2.51 | 189.00 | 60.80 | 61.20 | 10.30 |
| D26-R | 19.10 | 22.30 | 84.50 | 1.08 | 181.00 | 20.50 | 59.60 | 12.20 |
| D28-R | 11.70 | 4.72 | 79.50 | 0.40 | 198.00 | 28.00 | 5.67 | 11.60 |
| D29-R | 36.60 | 11.60 | 17.20 | 0.66 | 132.00 | 21.00 | 7.68 | 9.03 |
| D39-R | 6.22 | 1.88 | 137.00 | 0.96 | 213.00 | 58.90 | 31.80 | 12.30 |
| D40-R | 25.70 | 21.90 | 21.00 | 1.16 | 144.00 | 13.10 | 13.60 | 7.65 |
| D41-R | 42.50 | 25.30 | 28.90 | 0.88 | 155.00 | 41.50 | 24.00 | 7.73 |
| D43-R | 39.20 | 21.00 | 13.10 | 1.14 | 179.00 | 16.50 | 11.90 | 6.37 |
| D47-R | 33.80 | 10.50 | 12.80 | 0.77 | 114.00 | 24.80 | 5.71 | 5.51 |
| D48-R | 3.30 | 1.66 | 127.00 | 0.27 | 202.00 | 42.80 | 43.80 | 9.91 |
| D49-R | 54.90 | 17.70 | 25.40 | 1.43 | 155.00 | 61.00 | 45.00 | 4.84 |
| D50-R | 37.00 | 13.90 | 13.50 | 1.29 | 93.00 | 15.70 | 15.60 | 5.09 |
| D52-R | 43.50 | 20.10 | 23.10 | 2.41 | 146.00 | 57.30 | 27.00 | 9.76 |
| D55-R | 44.70 | 12.10 | 15.30 | 1.32 | 148.00 | 17.40 | 9.48 | 6.96 |
| D57-R | 30.10 | 17.70 | 24.20 | 0.75 | 172.00 | 7.90 | 5.36 | 10.30 |
| D61-R | 45.70 | 14.80 | 15.90 | 5.75 | 180.00 | 9.51 | 20.90 | 12.40 |
| D66-R | 25.30 | 27.10 | 39.50 | 0.83 | 158.00 | 23.90 | 22.90 | 10.60 |
| D68-R | 11.30 | 13.40 | 117.00 | 0.72 | 255.00 | 43.70 | 23.00 | 18.20 |
| D69-R | 19.90 | 11.30 | 21.10 | 0.78 | 122.00 | 8.57 | 4.57 | 8.08 |
| D72-R | 31.80 | 14.70 | 13.00 | 1.18 | 122.00 | 37.70 | 6.24 | 7.91 |
| D76-R | 32.90 | 11.40 | 14.60 | 1.61 | 114.00 | 39.10 | 8.30 | 5.57 |
| D81-R | 30.30 | 15.80 | 12.20 | 0.43 | 110.00 | 12.90 | 12.80 | 4.71 |
| D02-D | 88.00 | 14.80 | 32.90 | 1.31 | 270.00 | 78.90 | 12.50 | 0.00 |
| D03-D | 68.10 | 19.70 | 12.80 | 1.99 | 241.00 | 35.20 | 13.80 | 0.00 |
| D04-D | 63.30 | 14.10 | 16.70 | 2.42 | 247.00 | 31.30 | 13.90 | 0.00 |
| D05-D | 49.30 | 10.10 | 10.10 | 1.55 | 182.00 | 23.50 | 4.98 | 0.00 |
| D07-D | 147.00 | 23.50 | 59.60 | 1.91 | 323.00 | 228.00 | 41.70 | 0.00 |
| D09-D | 18.20 | 5.25 | 95.50 | 0.78 | 220.00 | 36.00 | 28.10 | 0.00 |
| D10-D | 10.50 | 5.45 | 122.00 | 0.72 | 261.00 | 47.10 | 29.10 | 0.00 |
| D11-D | 41.20 | 16.70 | 22.50 | 1.40 | 187.00 | 30.20 | 9.56 | 0.00 |
| D13-D | 15.30 | 5.09 | 92.30 | 0.68 | 227.00 | 38.00 | 32.40 | 0.00 |
| D14-D | 89.50 | 13.10 | 24.20 | 0.76 | 277.00 | 62.70 | 9.85 | 0.00 |
| D15-D | 35.00 | 15.40 | 15.20 | 1.97 | 234.00 | 19.90 | 9.49 | 0.00 |
| D16-D | 56.20 | 11.00 | 10.40 | 1.24 | 216.00 | 20.50 | 5.16 | 0.00 |

**Table A1.** *Cont.*

| Sample | Ca$^{2+}$ | Mg$^{2+}$ | Na$^+$ | K$^+$ | HCO$_3^-$ | SO$_4^{2-}$ | Cl$^-$ | CO$_3^{2-}$ |
|--------|-----------|-----------|--------|-------|-----------|-------------|--------|-------------|
| D17-D | 87.70 | 40.00 | 64.20 | 9.73 | 481.00 | 55.40 | 46.60 | 0.00 |
| D18-D | 54.00 | 10.70 | 16.50 | 0.96 | 206.00 | 18.00 | 7.43 | 0.00 |
| D19-D | 74.70 | 13.20 | 19.80 | 1.12 | 301.00 | 21.20 | 7.56 | 0.00 |
| D20-D | 67.20 | 10.70 | 12.30 | 0.93 | 214.00 | 26.30 | 7.62 | 0.00 |
| D21-D | 18.40 | 3.61 | 59.00 | 0.48 | 190.00 | 18.00 | 6.50 | 0.00 |
| D22-D | 36.00 | 9.47 | 34.30 | 0.57 | 192.00 | 14.30 | 5.08 | 0.00 |
| D23-D | 67.10 | 11.40 | 152.00 | 0.95 | 301.00 | 112.00 | 89.10 | 0.00 |
| D24-D | 21.80 | 9.04 | 131.00 | 1.04 | 294.00 | 55.50 | 37.60 | 0.00 |
| D25-D | 95.20 | 34.30 | 35.90 | 2.42 | 271.00 | 58.40 | 59.40 | 0.00 |
| D26-D | 58.30 | 29.00 | 92.90 | 1.10 | 333.00 | 38.20 | 79.10 | 0.00 |
| D28-D | 55.40 | 15.70 | 55.90 | 0.59 | 247.00 | 30.60 | 21.10 | 0.00 |
| D29-D | 29.80 | 8.63 | 38.30 | 0.61 | 183.00 | 14.60 | 7.88 | 0.00 |
| D39-D | 3.37 | 0.72 | 138.00 | 0.75 | 202.00 | 58.10 | 33.10 | 14.60 |
| D40-D | 56.40 | 21.90 | 22.70 | 0.84 | 231.00 | 22.00 | 18.60 | 0.00 |
| D41-D | 40.10 | 21.60 | 24.20 | 1.33 | 252.00 | 6.26 | 10.90 | 6.29 |
| D43-D | 52.00 | 16.60 | 14.20 | 1.39 | 246.00 | 9.42 | 9.79 | 0.00 |
| D47-D | 2.51 | 1.18 | 132.00 | 0.38 | 198.00 | 93.80 | 50.80 | 11.00 |
| D48-D | 119.00 | 27.60 | 39.50 | 1.80 | 330.00 | 102.00 | 104.00 | 0.00 |
| D49-D | 63.90 | 14.10 | 13.20 | 1.40 | 187.00 | 15.60 | 16.10 | 0.00 |
| D50-D | 69.50 | 15.90 | 19.20 | 2.45 | 254.00 | 49.20 | 15.90 | 0.00 |
| D52-D | 54.40 | 11.70 | 22.10 | 2.31 | 236.00 | 22.60 | 9.92 | 0.00 |
| D55-D | 40.20 | 17.40 | 24.00 | 0.78 | 226.00 | 8.02 | 5.74 | 0.00 |
| D57-D | 39.10 | 32.30 | 68.50 | 2.04 | 327.00 | 14.20 | 41.80 | 12.30 |
| D61-D | 49.90 | 32.10 | 42.00 | 4.38 | 300.00 | 71.60 | 19.70 | 0.00 |
| D66-D | 27.00 | 14.30 | 115.00 | 0.75 | 351.00 | 48.80 | 22.50 | 0.00 |
| D68-D | 88.00 | 17.90 | 19.30 | 0.52 | 321.00 | 35.50 | 15.60 | 0.00 |
| D69-D | 40.10 | 12.60 | 410.00 | 4.07 | 286.00 | 646.00 | 88.40 | 7.08 |
| D72-D | 58.30 | 11.10 | 16.70 | 1.48 | 221.00 | 31.70 | 8.00 | 0.00 |
| D76-D | 63.50 | 16.20 | 18.80 | 0.93 | 298.00 | 7.74 | 7.82 | 0.00 |
| D81-D | 135.00 | 20.70 | 834.00 | 4.72 | 219.00 | 1120.00 | 641.00 | 0.00 |

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
