# Peer review of "Hydrogeochemical Characterization and Its Seasonal Changes of Groundwater Based on Self-Organizing Maps"

_water, doi:10.3390/w13213065_

Round 1
Reviewer 1 Report
This is an interesting article about hydrogeochemical characteristics of the groundwater in the Hongjiannao Lake Basin.
The experiments were conducted with adequate methods and appropriate well-organized experimental design appropriate for examining the seasonal variability of the groundwater in the study area and its hydrochemical parameters.
The manuscript is very well written, with topical references and valuable results and conclusions. In my opinion, this study provides scientific basis for further investigations about the seasonal variability of the hydrogeochemical characteristics of groundwater in similar areas. Also, this study can be helpful to scientists in related fields.
Lines 596-653. I suggest the authors write in Bold the year for each reference.
Author Response
This is an interesting article about hydrogeochemical characteristics of the groundwater in the Hongjiannao Lake Basin.
The experiments were conducted with adequate methods and appropriate well-organized experimental design appropriate for examining the seasonal variability of the groundwater in the study area and its hydrochemical parameters.
The manuscript is very well written, with topical references and valuable results and conclusions. In my opinion, this study provides scientific basis for further investigations about the seasonal variability of the hydrogeochemical characteristics of groundwater in similar areas. Also, this study can be helpful to scientists in related fields.
Lines 596-653. I suggest the authors write in Bold the year for each reference.
Thanks to reviewer for their comments on this manuscript and the reviewer’s kindly suggestion. We have revised the year for each reference in the revised manuscript.

Reviewer 2 Report
The manuscript “water- 1441491” aims to study the groundwater bodies of Hongjiannao Basin by means several diagrams (e.g. Piper diagram, Gibbs diagram, Gaillardet diagram) and others methods (Chlor-alkali index, saturation index, and ion ratio). Moreover, based on self-organizing 14 maps (SOM), quantification error (QE), topological error (TE), and K-means algorithm, groundwater chemical data analysis was carried out to explore the seasonal variability.
I believe the manuscript should be published only after major revision.
Comments (P = page#/R = row#):
General comments
P3
The Figure 1 is not a hydrogeological map. I suggest to insert a geological map to make the text and succeeding results discussion consistent with the figure. Improve the discussion taking into account this aspect.
P3/R96
Add information about lithological and mineralogical point of view to make this section consistent with discussion .
P5/R181
Improve discussion in this section. The authors can take inspiration from these papers:
Zhang, B., Zhao, D., Zhou, P., Qu, S., Liao, F., & Wang, G. (2020). Hydrochemical characteristics of groundwater and dominant water–rock interactions in the Delingha Area, Qaidam Basin, Northwest China. Water, 12(3), 836.
Apollaro, C., Fuoco, I., Bloise, L., Calabrese, E., Marini, L., Vespasiano, G., & Muto, F. (2021). Geochemical Modeling of Water-Rock Interaction Processes in the Pollino National Park. Geofluids, 2021.
P7/R248
Please rewrite this section well-describing the two diagrams reported in Figure 4.
Moreover add “a” and “b” on the single plots in order to make clearer the discussion to the readers.
P8
Figure 4, write (a), (b) near the reference images;
P10/R304
Please add information about the used database in PHREEQC.
P15
Figure 9 shows concurrently SOMs in both rainy and dry season??Please add explanation in the text.
P 16
Figure 10 is not mentioned in the text. Please make sure all figures and tables are cited in the test.
P21
Add the following papers in the references list:
Zhang, B., Zhao, D., Zhou, P., Qu, S., Liao, F., & Wang, G. (2020). Hydrochemical characteristics of groundwater and dominant water–rock interactions in the Delingha Area, Qaidam Basin, Northwest China. Water, 12(3), 836.
Apollaro, C., Fuoco, I., Bloise, L., Calabrese, E., Marini, L., Vespasiano, G., & Muto, F. (2021). Geochemical Modeling of Water-Rock Interaction Processes in the Pollino National Park. Geofluids, 2021.
NOTE:
- A spell check in all text is required
Author Response

(The authors gave the same response as above.)

Round 2
Reviewer 2 Report
Concerning the previous comment “Please add information about the database used in PHREEQC”, I intended to add in the text the name of the thermodynamic database used by the PHREEQC code.
Please add this information in the text.
Author Response
Concerning the previous comment “Please add information about the database used in PHREEQC”, I intended to add in the text the name of the thermodynamic database used by the PHREEQC code.
Please add this information in the text.
Thanks to the reviewer’s kindly suggestion. We have added this information in the revised manuscript (in red).
The hydrogeochemical code PHREEQC is employed to conduct the calculation of SI speciation by using the database of phreeqc.dat.